# Mechanistic insight into carbon-carbon bond formation on cobalt under simulated Fischer-Tropsch synthesis conditions

C.J.(Kees-Jan) Weststrate [1*], Devyani Sharma [1,2], Daniel Garcia Rodriguez [1,2], Michael A. Gleeson [2], Hans O.A. Fredriksson[1] & J.W.(Hans) Niemantsverdriet[1,3]

Facile C-C bond formation is essential to the formation of long hydrocarbon chains in Fischer-Tropsch synthesis. Various chain growth mechanisms have been proposed previously, but spectroscopic identification of surface intermediates involved in C-C bond formation is scarce. We here show that the high CO coverage typical of Fischer-Tropsch synthesis affects the reaction pathways of $C_2H_x$ adsorbates on a Co(0001) model catalyst and promote C-C bond formation. In-situ high resolution x-ray photoelectron spectroscopy shows that a high CO coverage promotes transformation of $C_2H_x$ adsorbates into the ethylidyne form, which subsequently dimerizes to 2-butyne. The observed reaction sequence provides a mechanistic explanation for CO-induced ethylene dimerization on supported cobalt catalysts. For Fischer-Tropsch synthesis we propose that C-C bond formation on the close-packed terraces of a cobalt nanoparticle occurs via methylidyne (CH) insertion into long chain alkylidyne intermediates, the latter being stabilized by the high surface coverage under reaction conditions.

[1] SynCat@DIFFER, Syngaschem BV, De Zaale 20, 5612 AJ Eindhoven, The Netherlands. [2] Dutch Institute for Fundamental Energy Research (DIFFER), De Zaale 20, 5612 AJ Eindhoven, The Netherlands. [3] SynCat@Beijing, Synfuels China Technology Co. Ltd., Leyuan South Street II, No. 1, Huairou District, 101407 Beijing, China. *email: c.j.weststrate@syngaschem.com

Supported cobalt catalysts find their most widespread application in low-temperature Fischer–Tropsch synthesis (FTS), a process in which C-C bond-forming reactions produce long-chain hydrocarbon products from synthesis gas, a mixture of CO and $H_2$[1]. In today's fossil fuel-based economy, synthesis gas is predominantly manufactured from natural gas or coal, where FTS adds value by converting gaseous (gas-to-liquids) or solid (coal-to-liquids) reactants into more valuable products, such as food-grade wax, lubricants and sulfur-free transportation fuels. The FTS process will continue to play a role in future energy scenarios: synthesis gas can be derived from any carbon-containing source, e.g. biomass or even $CO_2$ may be used[2]. These renewable carbon sources offer an alternative route to produce a 'syncrude' that can, to a large part, replace petroleum as the principal feedstock of chemicals and the liquid fuels that power transportation modes (airplanes, ships, heavy vehicles) that cannot be readily replaced by fully electric alternatives. Insight into the molecular mechanism by which long-chain hydrocarbon species grow on the surface of the cobalt catalyst particle is of direct relevance to better understand the molecular origin of selectivity in FTS and may ultimately drive rational design of catalysts.

A large variety of chain growth mechanisms can be found in the literature, summarized in, e.g. refs. [3–5]. Growth intermediates of different chain length co-exist on the active surface, and steady-state isotopic transient kinetic analysis studies reveal that their concentration is low[6,7]. Furthermore, they are surrounded by much larger quantities of co-adsorbates such as $C_1H_{xad}$ species[8–10], $CO_{ad}$[7,9,10], $H_{ad}$[10] and long-chain products[11]. This complexity makes it impossible to distinguish those few active surface species from other adsorbates by, e.g. in situ infrared (IR) absorption spectroscopy[12]. We instead use a model approach to study $C_xH_y$ reactivity on a cobalt catalyst under conditions relevant to FTS. As shown hereafter, it is important to study how co-adsorbed $H_{ad}$ and CO affect $C_xH_y$ reactivity, since both adsorbates will be present on the surface under reaction conditions.

Water is a major by-product of FTS, and the high conversion levels reached lead to a water partial pressure that amounts to several bars during industrial operation. However, since chain growth also occurs under low conversion conditions where the $H_2O$ partial pressure is low, the presence of $H_2O$ does not appear to be essential to the chain growth mechanism and therefore it was omitted from our study. Moreover, surface science studies show that water adsorbs much weaker on Co(0001) than on both CO and hydrogen[13] and the $H_2O$ surface coverage under reaction conditions is expected to be low even when the $H_2O$ partial pressure is comparable to that of CO and $H_2$. An in situ X-ray absorption study of cobalt supported on a carbon nanofiber support shows that neither bulk oxidation nor substantial surface oxidation occurs on cobalt during FTS[14]. Furthermore, cobalt single crystals were found to be active for FTS[11,15–18], and the turnover frequencies reported are similar to those found for supported catalysts. This confirms that metallic cobalt is the active phase for chain growth and that insights from single crystal studies are of direct relevance for fundamental understanding of FTS.

We here use a Co(0001) model catalyst to study how $C_2H_{xad}$ species react to form a new C-C bond under FTS-like conditions, that is, in the presence of co-adsorbed hydrogen and $CO_{ad}$. Using this approach, we find that C-C bond formation is promoted by CO spectators, which stabilize the alkylidyne intermediate needed for this reaction. This finding can rationalize why CO promotes alkene dimerization on cobalt catalysts and reveals the hidden role of CO as promoter of chain growth during FTS on supported cobalt catalysts.

## Results

### $C_2H_4$ decomposition on Co(0001).

Synchrotron-based high-resolution X-ray photoemission spectroscopy (XPS) was used as the primary tool to determine both nature and concentration of the $C_xH_y$ surface intermediates that form at various stages in our experiments. The $C1s$ binding energy is sensitive to both chemical nature and binding site of the carbon atom, and excitation of the C-H stretch vibration along with the photoemission process gives rise to additional features at +350–400 meV (2800–3200 cm$^{-1}$) from the main photoemission peak[19,20], with an intensity that is proportional to the number of hydrogen atoms attached to the carbon atom that is photo-ionized[19,20].

Figure 1a shows high-resolution $C1s$ spectra of the different surface intermediates that form during heating of an ethylene-saturated surface in vacuum, previously discussed in detail elsewhere[21]. The two peaks at 283.9 and 283.4 eV in the spectrum at 100 K are attributed to the two carbon atoms of ethylene[21,22]. The changes seen around 180 K in the heat map of $C1s$ spectra recorded during heating (Fig. 1b) are attributed to a combination of ethylene desorption and decomposition, the latter producing 0.12 monolayer (ML) acetylene ($C_2H_{2ad}$) + 0.24 ML $H_{ad}$. The $C1s$ spectrum of adsorbed acetylene shows a peak at 283.3 eV, which accounts for both carbon atoms and is accompanied by two small shoulders at +0.37 and +0.74 eV due excitation (double excitation) of the C-H stretching vibration[21–23]. Acetylene remains stable up to 400 K where it dehydrogenates completely (see $H_2$ desorption data reported below). Atomic carbon is the only adsorbate present after heating to 630 K, as evident from the $C1s$ peak at 282.8 eV[24].

### The influence of CO spectators on $C_2H_x$ reactivity.

The mixed $C_2H_{2ad}/H_{ad}$ layer produced by heating the ethylene-covered surface to 220 K was chosen as the starting point to investigate how CO spectators affect $C_2H_{xad}$ reactivity. A top view of the $C1s$ spectra (Fig. 1c) shows that acetylene gets converted around 270 K when heated in the presence of $1 \times 10^{-7}$ mbar CO. A second reaction step is seen around 310 K, while extensive dehydrogenation occurs above 350 K to produce a mixture of 'polymeric'[24] (284.4 eV) and atomic carbon (282.7 eV) at 630 K. This is different from the carbon layer found after heating in vacuum, which consisted exclusively of atomic carbon (Fig. 1a and Supplementary Note 1).

High-resolution $C1s$ spectra reveal the identity of the intermediates formed at each stage. Figure 1d only shows the spectral features due to the dominant product to simplify the discussion, while a detailed discussion of the as-measured spectra is provided in Supplementary Note 2. The spectrum recorded after heating to 200 K shows the spectral shape of acetylene, and co-adsorption of CO causes only a slight broadening of the acetylene peak. We find that up to 0.30 ML CO can adsorb alongside the 0.12 ML acetylene (and 0.24 ML $H_{ad}$), with 0.14 ML $CO_{ad}$ residing on top sites (285.7 eV[25]) and 0.16 ML in threefold hollow sites (285.3 eV[25]).

Acetylene gets converted to ethylidyne between 220 and 270 K, as evident from two new peaks at 282.9 and 283.5 eV after heating to 285 K. The peak at 282.9 eV does not show any loss features related to the presence of C-H bonds, in line with the assignment to the surface-bound carbon atom of ethylidyne ($\equiv\underline{C}$-$CH_3$)[21,26]. The three hydrogen atoms in the methyl group of ethylidyne give rise to prominent vibrational signals at +0.4 and +0.8 eV relative to the main peak at 283.5 eV. The $C1s$ binding energy of the methyl group is 0.5 eV lower than the typical value reported for methyl groups in a variety of other $C_xH_{yad}$ adsorbates[19,21,27]. This can be attributed to the presence of $CO_{ad}$: a similar downward shift of the methyl group binding energy of ethylidyne adsorbed

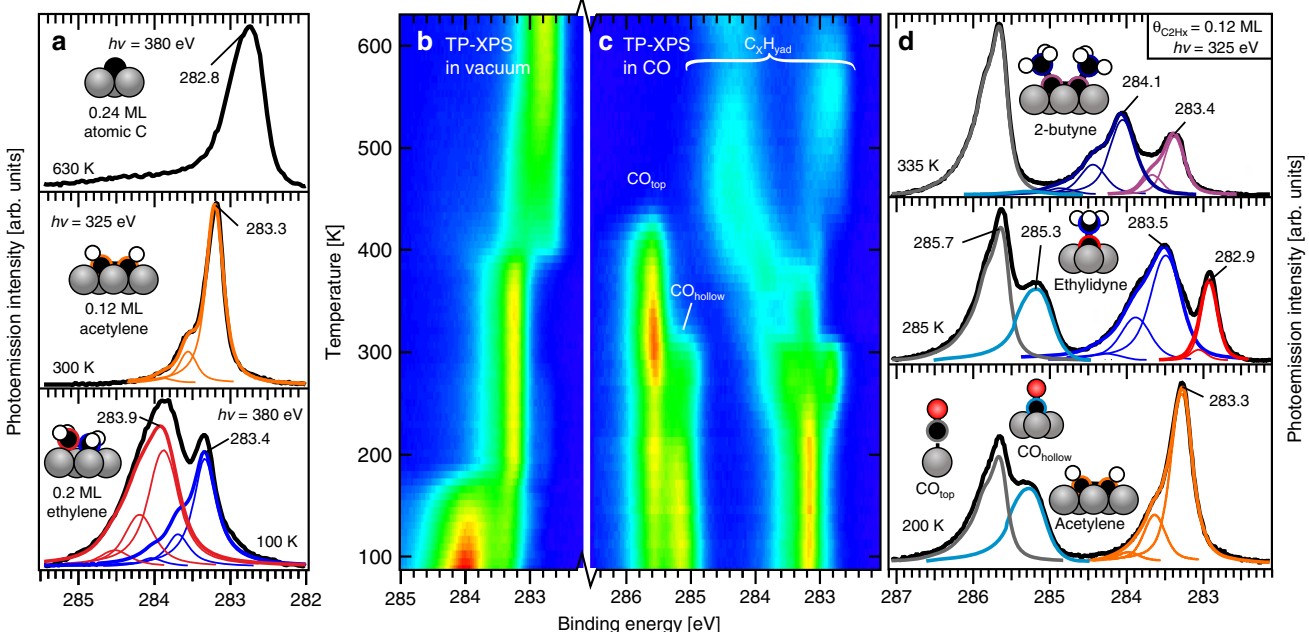

**Fig. 1 C1s spectra of C$_x$H$_y$ adsorbates at different conditions. a** High-resolution spectra after heating ethylene-saturated Co(0001) in vacuum to the indicated temperatures. **b** Heat map of C1s spectra recorded during heating of ethylene-covered Co(0001) in vacuum. **c** Heat map during heating of the C$_2$H$_{2ad}$/2H$_{ad}$-covered surface in the presence of $1\times10^{-7}$ mbar CO (h$\nu$ = 380 eV, 7 K per spectrum, low intensity = blue, high intensity = red, heating rate 0.2 K s$^{-1}$). **d** High-resolution spectra after heating C$_2$H$_{2ad}$/2H$_{ad}$ in CO to the temperatures indicated in the figure.

on Rh(111) upon co-adsorption of CO was reported previously[26], and our reference experiment (Supplementary Note 3) showed that co-adsorption of CO alongside adsorbed propyne causes a CO coverage-dependent shift of the methyl (H$_3$C̲-C≡CH) binding energy, from 284 eV down to 283.5 eV for the highest CO coverage. The CO coverage at this point is 0.30 ML but now with 0.18 ML CO adsorbed on-top and 0.12 ML in hollow sites.

Two ethylidyne adsorbates couple around 310 K to form a 2-butyne product. This proposition is confirmed by the high-resolution spectrum recorded after heating to 335 K, which closely resembles the previously reported high-resolution spectrum of 2-butyne adsorbed on Ni(111)[28]. The CO coverage has dropped to 0.21 ML, and only the top sites are populated at this point. Reference experiments using propyne (Supplementary Note 3)[21] show that CO$_{top}$ has only a minor influence on the methyl binding energy, and both methyl groups in 2-butyne, H$_3$C̲-C≡C-C̲H$_3$, therefore appear at 284.1 eV, the typical value for -C̲H$_3$[19,21,27]. The 283.4 eV peak is attributed to the two central carbon atoms of adsorbed 2-butyne, H$_3$C-C̲≡C̲-CH$_3$. Their binding energy value is identical to that of the central carbon atom of adsorbed propyne H$_3$C-C̲≡CH, which is unsurprising since the adsorption site and immediate surroundings of the central carbon atoms in adsorbed propyne and 2-butyne are practically identical.

These results thus show that CO spectators cause acetylene to react with surface hydrogen to produce ethylidyne around 270 K. Two ethylidyne products subsequently couple around 310 K to produce adsorbed 2-butyne. Other experimental techniques corroborate these conclusions and reveal additional details. Temperature-programmed reaction spectroscopy (TRPS; Fig. 2a) shows that ~0.08 ML ethylene desorbs intact during heating of the ethylene-saturated surface in vacuum. Hydrogen desorbs in two steps, with a peak at 320 K due to recombination of the 0.24 ML H$_{ad}$ produced by ethylene decomposition to acetylene and another peak around 400 K, which amounts to 0.24 ML H$_2$ and is attributed to complete dehydrogenation of acetylene[21,29]. The presence of CO strongly affects the H$_2$ TPRS trace as shown in

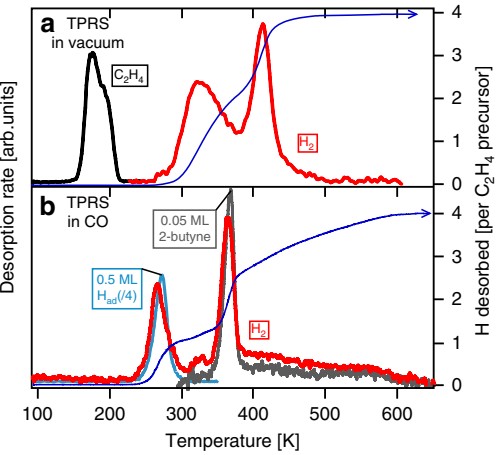

**Fig. 2 Temperature-programmed reaction spectroscopy.** Ethylene (black line) and hydrogen desorption (red line) during heating (0.2 K s$^{-1}$) of **a** an ethylene-saturated Co(0001) in vacuum and **b** a 0.12 ML C$_2$H$_{2ad}$/0.24 ML H$_{ad}$-covered surface in $1\times10^{-7}$ mbar CO. H$_2$ desorption from a 0.5 ML H$_{ad}$-covered surface heated in CO (light blue and divided by 4 to facilitate comparison) and H$_2$ desorption due to 2-butyne decomposition (grey line, in $1\times10^{-7}$ mbar CO) are added for reference. The dark blue lines show the integral of the H$_2$ desorption traces in the absence and presence of CO, normalized to the number of H atoms in the ethylene precursor.

Fig. 2b. The peak around 270 K is attributed to desorption of surface-bound hydrogen, shifted downward due to the presence of CO[30–32]. The peak area indicates that 0.12 ML H$_{ad}$ desorbs around 270 K, only 50% of the 0.24 ML H$_{ad}$ that was present prior to CO exposure. This is explained by acetylene hydrogenation to ethylidyne (HC≡CH + H$_{ad}$ → ≡C-CH$_3$), a reaction that consumes one H$_{ad}$ for each acetylene adsorbate converted. The H$_2$ desorption trace above 320 K has a close similarity to that of a reference experiment in which 0.05 ML 2-butyne was heated in the presence of CO. This supports the conclusion that adsorbed

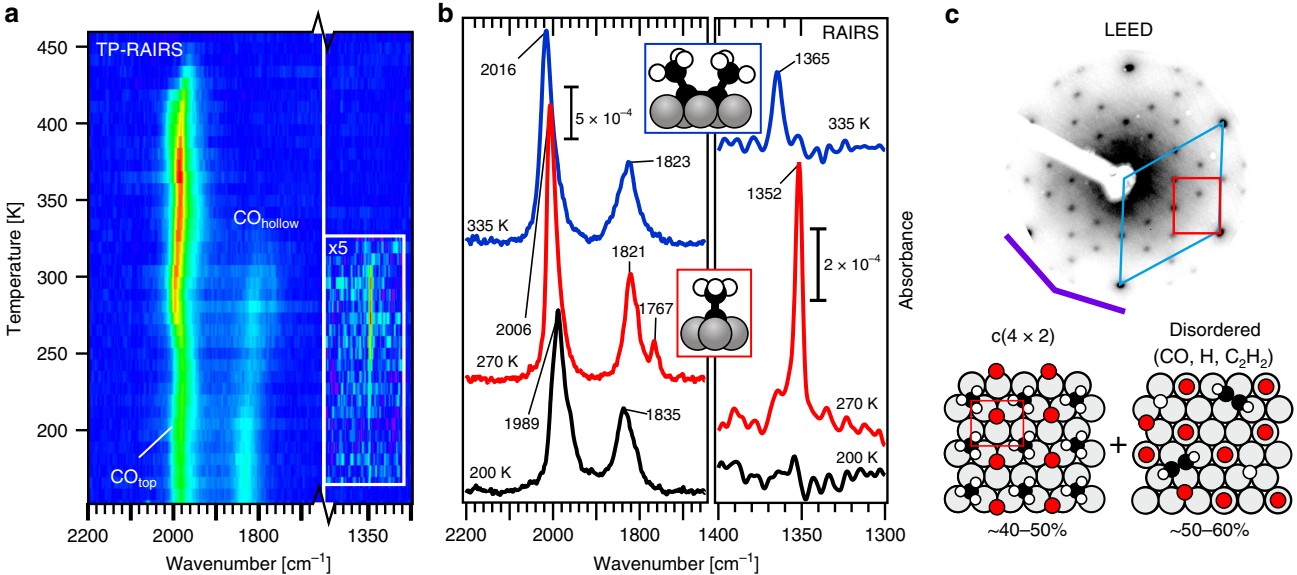

**Fig. 3 IR absorption spectroscopy and electron diffraction. a** TP-RAIRS (8 K per spectrum, low absorbance = blue, high absorbance = red) and **b** IR absorption spectra ($T_{sample}$ = 90 K) during heating of $C_2H_{2ad}$/$2H_{ad}$ adsorbed on Co(0001) in CO. The symmetric methyl bending mode (1352 cm$^{-1}$) indicate that ethylidyne is present between 250 and 300 K. **c** LEED (80 eV) shows a c(4 × 2) pattern between 270 and 305 K, attributed to islands of an ordered ethylidyne/CO layer that cover up to 50% of the surface ($p_{CO}$ = 1 × 10$^{-7}$ mbar, heating rate 0.2 K s$^{-1}$, $\theta_{C2Hx}$ = 0.12 ML).

2-butyne is produced by subsequent ethylidyne coupling (Supplementary Note 4). We attribute the pronounced peak at 370 K to dehydrogenation of the two methyl groups in 2-butyne[21]. The remaining hydrogen gradually leaves the surface between 380 and 600 K, characteristic of dehydrogenation of 'polymeric' surface carbon, the formation of which was also evident from XPS. The small H$_2$ desorption peak around 320 K is attributed to a minor quantity of ethylidyne that did not find a coupling partner and instead dehydrogenate back to acetylene.

Figure 3a shows the top view of a series of reflection absorption infrared spectra (RAIRS), recorded during heating of an $C_2H_{2ad}$/2 H$_{ad}$-covered surface in CO. Individual high-quality spectra at the temperatures indicated are shown in Fig. 3b. The strong bands around 2000 and 1840 cm$^{-1}$ are due to CO$_{top}$ and CO$_{hollow}$, respectively[25]. CO adsorbed alongside small $C_xH_y$ adsorbates attenuates absorption bands in the C-H (~2800–3000 cm$^{-1}$) and C-C (1000–1200 cm$^{-1}$) stretching regions, a phenomenon that has been observed before for ethylidyne adsorbed alongside CO on, e.g. Ru(0001) and Pt(111)[32–34]. This leaves only the C-H bending region (1500–1300 cm$^{-1}$) as our main source of information. The 1352 cm$^{-1}$ band, which appears around 220 K and disappears again around 300 K, can be readily assigned to the symmetric bending mode ($\delta_s$-CH$_3$) of ethylidyne[35]. The band at 1767 cm$^{-1}$ that appears and disappears together with the 1352 cm$^{-1}$ band has been reported previously for CO$_{hollow}$ co-adsorbed with ethylidyne on both Rh(111) and Ru(0001)[34,36].

The ethylidyne product forms an ordered adsorbate overlayer together with CO, as evident from the c(4 × 2) diffraction pattern that was found between 270 and 305 K by low-energy electron diffraction (LEED) (Fig. 3c). The ordered ethylidyne/CO$_{hollow}$ overlayer that causes this pattern is known from earlier studies on Rh(111)[26,36]. The local ethylidyne coverage in the c(4 × 2) structure is 0.25 ML. However, the overall $C_2H_{xad}$ coverage in our experiment is only 0.12 ML. This means that the c(4 × 2) pattern is caused by islands of ethylidyne/CO$_{hollow}$ that cover only 40–50% of the surface, leaving the other half covered by a disordered layer, which contains CO as well as minor quantities of unreacted acetylene and H$_{ad}$. A more detailed analysis of this structure is provided in Supplementary Note 5. The c(4 × 2)

pattern disappears above 300 K, and the IR absorption spectrum after heating to 335 K shows a weak band at 1365 cm$^{-1}$. This is attributed to the $\delta_s$-CH$_3$ mode of adsorbed 2-butyne, an assignment that is supported by reference spectra using 2-butyne (Supplementary Note 4).

**XPS at near-ambient pressure (NAP).** The reactivity of acetylene under high-coverage conditions was further explored using the HIPPIE beamline of MAX IV, which allows XPS measurements at near-ambient pressures. Figure 4 shows a series of C1s spectra recorded during an isothermal experiment at 313 K in which a 0.08 ML acetylene-covered Co(0001) surface (prepared by dosing ethylene at 313 K) was exposed to an increasingly high H$_2$ pressure. A H$_2$ pressure of ~10$^{-5}$ mbar would be sufficient to create a high hydrogen coverage at 313 K[31,37], but the spectrum shape shows that co-adsorbed hydrogen did not cause the acetylene to react.

A CO contamination in the gas led to the appearance of CO$_{ad}$ when the hydrogen pressure was increased to 1 × 10$^{-2}$ mbar. The appearance of CO$_{ad}$ coincides with the conversion of acetylene to ethylidyne, the temporary formation of the latter being evident from the transient peaks at 282.9 and 283.8 eV. The CO$_{hollow}$ peak (285.2 eV) comes and goes together with the ethylidyne-related peaks, consistent with the RAIRS results which show that CO$_{hollow}$ is associated with ethylidyne. Ethylidyne reacts further to produce 2-butyne, evident from a peak at 283.4 eV due to the two central carbon atoms and at 284.1 eV due to the two methyl groups. Acetylene conversion reached completion after 400 s in 1 × 10$^{-2}$ mbar H$_2$, and a pressure increase to 1 × 10$^{-1}$ mbar H$_2$ did not affect 2-butyne at all. Interestingly, this pressure increase triggers the desorption of CO$_{ad}$. We attribute this to the same repulsive interactions that cause the decreased H$_2$ desorption temperature when H$_{ad}$ is heated in CO (Fig. 2b). With an overwhelming majority of hydrogen in the NAP experiment, H$_{ad}$-induced destabilization of CO$_{ad}$ causes it to desorb at 313 K, significantly below its 'normal' desorption temperature of 350–400 K[25].

The invariance of the C1s spectral shape during subsequent heating in 1 × 10$^{-1}$ mbar H$_2$ (Fig. 5) indicates that 2-butyne

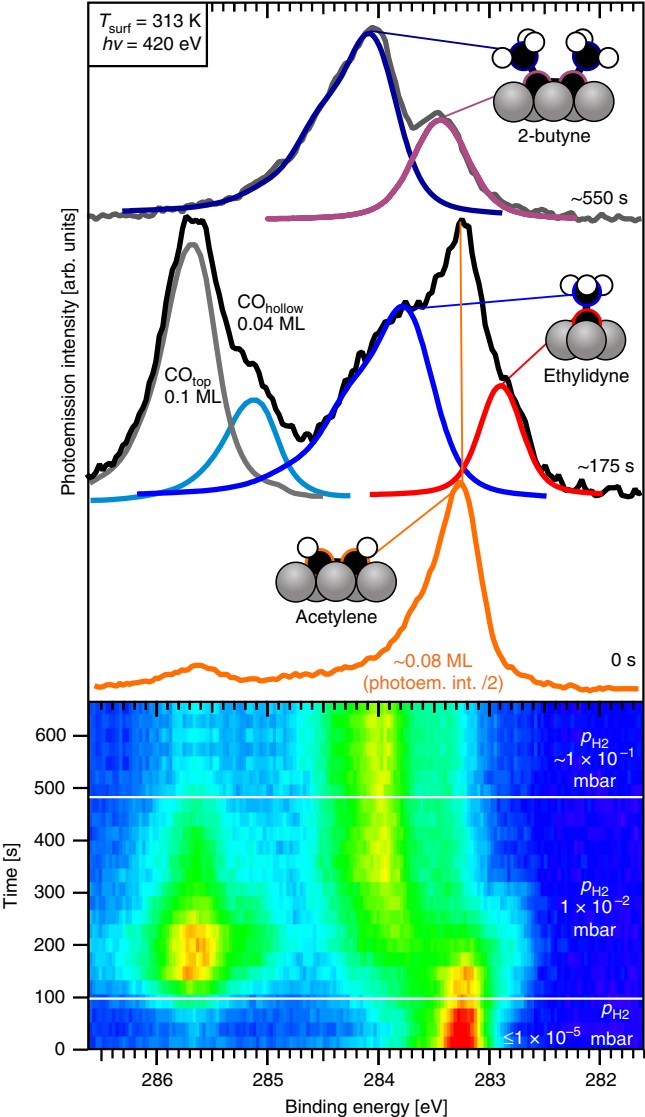

**Fig. 4 XPS at near-ambient pressures.** C*1s* spectra recorded during exposure of an acetylene-covered Co(0001) surface to increasingly high $H_2$ pressures at $T = 313$ K. The time evolution of the C*1s* spectra is shown in the lower panel, whereas the top view shows the spectra at specific stages of the experiment ($h\nu = 420$ eV). Note that the signal intensity of the acetylene spectrum in the upper panel was divided by two to facilitate comparison.

remains intact during heating. The decrease of the signal intensity >370 K is attributed to loss of 2-butyne from the surface. Since computed alkyne adsorption energies are in excess of 200 kJ mol$^{-1}$, the possibility that 2-butyne simply desorbs can be excluded[38]. We instead propose that 2-butyne is hydrogenated to 2-butene, which desorbs upon formation. A simple kinetic analysis (Supplementary Note 6) yields an apparent activation energy of $106 \pm 12$ kJ mol$^{-1}$ for 2-butyne hydrogenation.

## Discussion

The results presented here provide uniquely detailed information on the mechanism by which C-C bonds are formed between two $C_xH_y$ species adsorbed on a Co(0001) model catalyst in the presence of co-adsorbed CO and $H_{ad}$. After discussing the barriers for ethylidyne formation and ethylidyne dimerization and the possible origins of the driving force of these reactions, we

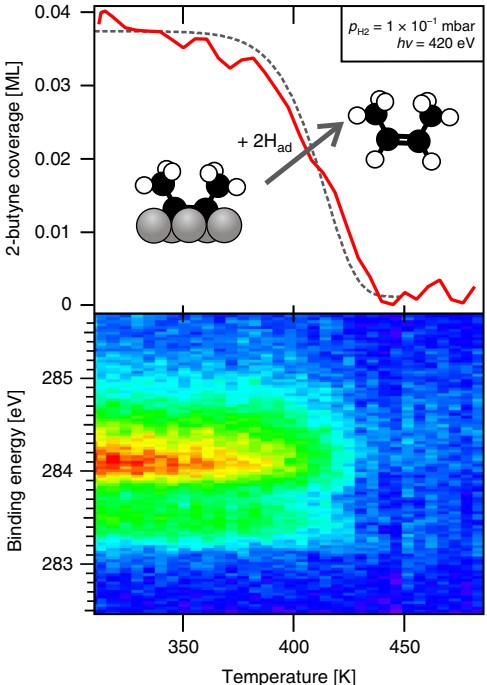

**Fig. 5 XPS during heating in a hydrogen atmosphere.** Peak area and top view of the C*1s* spectra recorded during heating of (~0.04 ML) 2-butyne-covered Co(0001) in the presence of $1 \times 10^{-1}$ mbar $H_2$ (0.2 K s$^{-1}$, $h\nu = 420$ eV). The experimental data is shown by a solid red line, while the output of our kinetic model is shown as a dashed grey line.

continue with a discussion of how these findings relate to C-C bond formation reactions catalysed by metallic cobalt in the presence of CO.

Quantitative analysis of the TP-XPS data (Supplementary Note 5) yields the coverage of all C-containing adsorbates during the CO-induced reaction while $\theta_H$ can be deduced from the $H_2$ desorption trace. Figure 6 shows the adsorbate coverages together with the output of a simple mean field microkinetic model (Supplementary Note 6) used to estimate reaction barriers for acetylene hydrogenation and ethylidyne dimerization. The conversion of acetylene to ethylidyne requires the addition of two hydrogen atoms to one end of the molecule and abstraction of a hydrogen atom from the other end. Our analysis yields an overall barrier height of only $60 \pm 6$ kJ mol$^{-1}$, far below the computed barriers >100 kJ mol$^{-1}$ as predicted by density functional theory starting with either acetylene hydrogenation or acetylene dehydrogenation as a first step[39–41]. This indicates that the high CO coverage may also lower the reaction barriers involved in the conversion of acetylene.

Acetylene is the most stable $C_2H_x$ adsorbate in the absence of CO[21,38–41], and it is therefore the sole product of ethene decomposition around 180 K. The experiments show that CO causes conversion of acetylene to ethylidyne, and the driving force for this must be that ethylidyne becomes more stable than acetylene when CO is present on the surface[32,40]. Two experimental observations give more insight into why ethylidyne is preferred when CO is present: (i) repulsive interactions between $H_{ad}$ and $CO_{ad}$ destabilize $H_{ad}$ (Fig. 2b), thereby destabilizing one of the reactants. (ii) The formation of ordered $CO_{hollow}$/ethylidyne layer islands with a local high coverage is driven by attractive interactions, indicating that the ethylidyne reaction product is stabilized by $CO_{ad}$ (Fig. 3c). Mate et al.[42] attributed the ordering of mixed ethylidyne/CO layers on Rh(111) to favourable dipole–dipole interactions due to the oppositely oriented dipoles

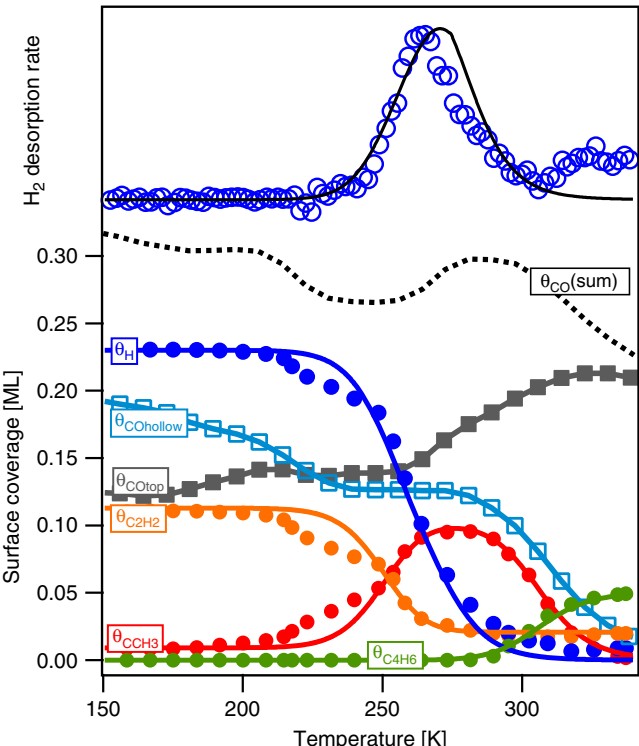

**Fig. 6 Surface coverage during heating in CO.** Markers show the surface coverages obtained from analysis of TP-XPS and TPRS during heating of a $C_2H_{2ad}/2H_{ad}$-covered Co(0001) surface in $1 \times 10^{-7}$ mbar CO (0.2 K s$^{-1}$). The output of a mean field microkinetic model is provided by the solid lines. The measured and simulated hydrogen desorption spectrum is added for comparison.

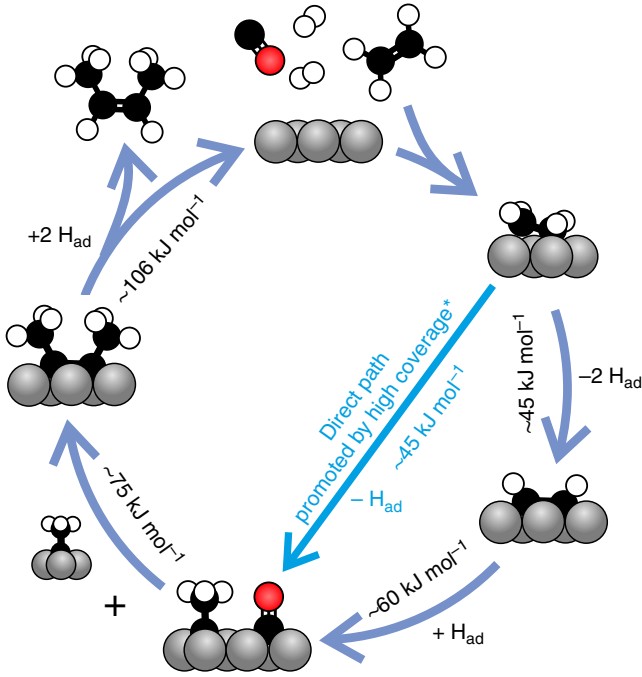

**Fig. 7 Proposed catalytic cycle of CO-induced ethylene dimerization.** The proposal is based on the experimental findings on Co(0001) presented here and in ref. [21*].

of CO and ethylidyne. A third contributing factor is that a high surface coverage favours species that occupy less space on the surface[21]. This makes acetylene hydrogenation to ethylidyne favourable, first because it incorporates a surface hydrogen into the $C_xH_y$ adsorbate, and second, the footprint of ethylidyne is expected to be lower than that of acetylene since the former adsorbs through only one carbon atom.

Earlier studies have shown that methylidyne dimerizes around 250 K on both Ni(111) and Co(0001) to form acetylene[31,43,44]. We here show that ethylidyne, the methyl-substituted analogue of methylidyne, reacts in a similar way to produce 2-butyne. We attribute the comparatively higher temperature (310 vs 250 K) to increased steric hindrance in the transition state for coupling when hydrogen is replaced by a bulky methyl group. Ethylidyne reactivity in the absence of CO was studied by intentionally maximizing beam-induced ethylene decomposition (Supplementary Note 7). Instead of dimerization, we find that ethylidyne dehydrogenates to acetylene around 280 K during heating in vacuum[21]. Since dimerization of $CH_{ad}$ does not require CO to be present, we propose that $CO_{ad}$ is not needed for the C–C bond formation reaction itself. Instead, CO facilitates the C–C bond-forming reaction indirectly: it stabilizes ethylidyne so that it still exists at 310 K in our temperature-programmed experiment, the temperature required to overcome the barrier of $75 \pm 7$ kJ mol$^{-1}$ associated with ethylidyne dimerization.

The work of Eidus et al.[45] shows that traces of CO during ethylene hydrogenation on a supported cobalt catalyst changes the product selectivity, from ethane to $C_4$ products. Cant et al.[46] reproduced these early studies and reported cis-2-butene as the main $C_4$ product of ethylene hydrogenation at 393 K in presence of CO. CO-induced stabilization of ethylidyne is key to

understand why CO promotes ethylene dimerization on cobalt. Figure 7 shows the proposed catalytic cycle for this reaction, with activation energies derived from direct observation of the respective reactions on our single-crystal model catalyst. Ethylene decomposition initially produces acetylene, which is quickly hydrogenated to ethylidyne under the influence of CO spectators. Alternatively, a more direct pathway may exist from ethylene to ethylidyne: our previous work shows that ethylidyne can also form directly as a minor product of ethylene decomposition around 180 K, but only when the surface is highly covered by $C_xH_{yad}$ and/or $H_{ad}$[21]. The C–C bond is then formed via coupling of two ethylidynes. Hydrogenation of the alkyne coupling product is the slowest step, and the 2-butyne concentration under reaction conditions is therefore expected to be rather high. This facilitates the formation of carbonaceous deposits via side reactions such as 2-butyne dehydrogenation and alkyne cyclo-trimerization[38,47], and explains the fast catalyst deactivation which was attributed to carbon deposition by Cant et al.[46].

The surface composition during CO-promoted ethylene dimerization is similar to the situation during FTS, as in both cases hydrocarbon surface intermediates react to form new C–C bonds on a cobalt surface covered with CO and surface hydrogen. For FTS, Bezemer et al.[14] report a CO turnover frequency of only $2.3 \times 10^{-2}$ s$^{-1}$ for their most active cobalt catalyst (tested at 483 K and 35 bar). This translates to a production rate of only 40 monomers per second on each cobalt crystallite (hemispherical, average diameter 8.5 nm). So even in the unlikely case that all monomers insert into a single chain, it still takes ~0.25 s to grow a one $C_{10}$ product molecule per particle ($C_{10}$ being the average chain length when $\alpha = 0.9$[4]). Such slow growth on the second time scale can only involve very stable growth intermediates that have difficulty leaving the surface. Our work shows that alkynes and alkylidynes are indeed very stable, and the NAP-XPS experiment (Fig. 5) confirms that alkyne hydrogenation is a slow reaction that only proceeds at a significant rate above 370 K. Their high stability and long lifetimes thus make alkynes and

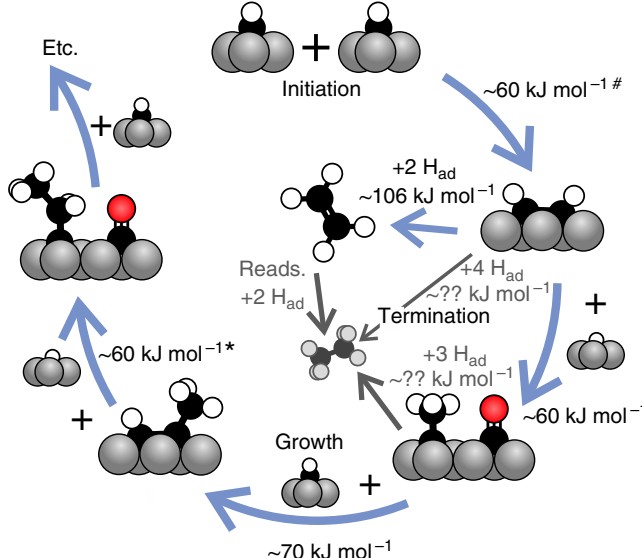

**Fig. 8 Proposed chain growth mechanism on close-packed terraces.** Steps depicted in colour were either observed directly or derived from direct observation of analogue reactions. The barrier for CH + CCH$_2$R is proposed to be between the 60 kJ mol$^{-1}$ reported for CH dimerization[4,31,43,44]# and 75 kJ mol$^{-1}$ for ethylidyne dimerization. Chain termination via hydrogenation can occur at any point but is only shown explicitly for C$_2$H$_x$. Experimental proof for CO-induced propylidyne and 1-butylidyne formation can be found in ref. [32]*.

alkylidynes feasible intermediates for chain growth under FTS conditions.

Our experiments moreover show that CO spectators stabilize alkylidyne over all other forms of C$_x$H$_{yad}$ and that alkylidynes are highly reactive in C-C bond-forming reactions. These findings provide compelling support for the previously proposed alkylidyne chain growth mechanism[4,38], schematically depicted in Fig. 8. New chains are initiated by coupling of two methylidyne (CH) monomers, the most stable form of CH$_{xad}$, both without[31,48,49] and with CO co-adsorbed[41]. Subsequent, CO-promoted hydrogenation of acetylene produces the ethylidyne needed for further growth. With the C$_1$H$_x$ concentration being much higher than that of growing chains under FTS conditions[6,8], methylidyne insertion (producing adsorbed pro-pyne) prevails over coupling with ethylidyne that we find in our model experiments. In analogy to the acetylene–ethylidyne reaction, CO also promotes conversion of the propyne coupling product to the propylidyne form[32] needed for further growth. Chain termination requires hydrogenation and is comparatively slow, as illustrated by the difficulty to hydrogenate 2-butyne in our experiments. The mechanism observed in our experiments is of the carbide type, and although CO is promoting chain growth by stabilizing the intermediates in the correct form, it does not participate in the C-C bond-forming reaction. A detailed analysis of the O$1s$ spectra provided in Supplementary Note 8 shows that the formation of oxygen-containing intermediates during the experiments can be excluded.

The experimentally observed surface reactions presented here show that C-C bond formation via alkylidyne coupling is favourable on the close-packed terraces that constitute ~60% of the surface of an (fcc-)Co nanoparticle[4,41]. The methylidyne monomers required for chain growth during FTS can either be supplied via H-assisted routes on the terrace or via spill-over from adjacent step sites that are active for CO dissociation[18,50] and surround the nm-sized close-packed terrace exposed by an fcc-Co nanoparticle[4]. An fcc-Co nanoparticle also exposes (100) facets as

well as step and kink sites[51], and the role of the different structural elements remains a topic of debate[52]. Given that our experiments only speak about C$_x$H$_y$ reactivity on close-packed terraces, we can only provide a qualitative consideration about chain growth on other sites. Since the relative stabilities of C$_x$H$_y$ adsorbates depends on the surface structure, chain growth may proceed via a different pathway on other parts of the catalyst surface. In fact, experiments on stepped Ni(111) show that step sites favour C-C bond scission rather than C-C bond formation[53,54], and C-C bond scission was reported on Ni(100) at temperatures below 300 K[55]. This is in line with computational work on cobalt, which shows that 2 C$_1$H$_x$ adsorbates are more stable than C$_2$H$_x$ adsorbates on both Co(100) and stepped Co (211), while acetylene and ethylidyne are instead significantly more stable than 2 CH species on Co(111)[41]. In other words, C-C bond scission is favoured on steps and (100) facets, whereas C-C bond formation is favoured on terrace sites. This means that growth of long chains at step edges can only occur when exo-thermic C-C bond breaking is strongly suppressed, e.g. by a local high coverage at the step site. The concentration of carbon is indeed expected to be high on step sites due to their high affinity for carbon[41,56]. Chain growth requires growing chains to physi-cally meet with a monomer, and since the growing chain con-centration is expected to be small[6], the nearest monomer is most probably generated at some distance from the growth inter-mediate. A high local coverage at step edges poses a significant barrier for C$_x$H$_y$ diffusion along the one-dimensional step edge, making it difficult for a monomer to reach a growing chain that is located a few sites away. Thus, owing to the limited mobility along the highly covered step sites, only those monomers formed at a small ensemble of step sites will contribute to a single chain, whereas monomers produced at step sites far away may not be able to reach the growing chain and make methane instead. Instead, spill-over of monomers to the adjacent (two-dimen-sional) terrace circumvents blockages of strongly adsorbed species at step sites so that monomers and growing chains can more easily diffuse over a larger distance and monomers produced by different active sites located at the edges of a single terrace can all be incorporated into the same product chain.

In summary, our detailed investigation of the reactivity of C$_2$H$_{xad}$ species on Co(0001), both at low and near-ambient reactant pressures, reveals that a high coverage of CO$_{ad}$ spectators profoundly influences the reactivity of hydrocarbon adsorbates. CO-induced hydrogenation of adsorbed acetylene-producing ethylidyne ($\equiv$C-CH$_3$) is a facile reaction that occurs around 270 K. The driving force for the reaction is provided by a com-bination of CO-induced destabilization of the H$_{ad}$ reactant and stabilization of the ethylidyne product. Formation of 2-butyne (H$_3$C-C$\equiv$C-CH$_3$) occurs around 310 K via ethylidyne dimeriza-tion and highlights the high reactivity of alkylidynes ($\equiv$C-CH$_2$R) for C-C bond formation on the close-packed surface of cobalt. The finding that CO$_{ad}$ stabilizes C$_x$H$_{yad}$ adsorbates in the alky-lidyne form, which readily forms a new C-C bond with other alkylidynes, rationalizes why CO promotes ethylene dimerization on cobalt catalysts. For FTS, we propose that C-C bonds form on the close-packed facets via coupling of a long-chain alkylidyne with a methylidyne monomer to form a 1-alkyne adsorbate. CO spectators promote chain growth by stabilizing the growing chain in the alkylidyne form needed for C-C bond formation. NAP-XPS shows that 2-butyne hydrogenation is slow. This suggest that the alkyne and alkylidyne adsorbates terminate only slowly, an essential requirement for the growth of long chains[5].

## Methods

**Sample description and sample cleaning.** The disc-shaped ($d = 8$ mm) Co(0001) single crystal (Surface Preparation Laboratory) was cleaned by cycles of ~10 min

sputtering (1 kV Ar$^+$) at 650 K followed by ~10 min annealing at the same temperature[21,32]. Residual carbon was most efficiently removed by dosing ~1× $10^{-7}$ mbar O$_2$ at 650 K for a few minutes. The excess surface oxygen can be removed by a short sputtering step or by exposure to H$_2$ ($10^{-6}$–$10^{-5}$ mbar) at 650 K. In this way, the carbon concentration could be reduced to below the sensitivity of synchrotron XPS.

**Synchrotron XPS at ultra-high vacuum (UHV) condition.** The synchrotron XPS measurements under UHV conditions reported here were performed at the SuperESCA beamline of ELETTRA, the European synchrotron light source located in Trieste, Italy[57]. For these experiments, the sample was spotwelded to a Ta rod, which was in direct thermal contact with a liquid nitrogen reservoir such that a sample temperature of 80 K could be reached. The sample was heated by the radiation of three tungsten filaments located close to its backside. Sample temperatures were measured using a K-type thermocouple, spotwelded to the side of the sample. Binding energies are reported relative to the Fermi edge, which was remeasured after each change of the photon energy. CO (SIAD, 99.95%), propene (Messer, 99.5%) and ethylene (SIAD, 99.995%) were used without further purification during the experiments at the SuperESCA beamline. XPS did not show any indication of contaminations introduced by dosing these gases. The ordered layer formed by CO after adsorption at room temperature, with a coverage of 0.33 ML and a ($\sqrt{3}\times\sqrt{3}$)R30° in LEED, was used as a reference point for both C1s and O1s quantification[21,25]. C1s and O1s spectra were recorded with 800 and 1100 eV for this known structure, where the higher photon energies were chosen to minimize the impact of photoelectron diffraction effects (Supplementary Note 2) on the signal intensity. The concentration of C$_x$H$_y$ species derived from XPS was found to be in good agreement with the values derived from TPRS, as discussed hereafter.

**Synchrotron XPS at near-ambient conditions.** The NAP XPS measurements were performed at the HIPPIE beamline of MAX IV, Lund, Sweden. The system consists of multiple chambers separated by gate valves. For the results presented here, only the preparation chamber, the transfer chamber (both with a base pressure of ~5× $10^{-10}$ mbar) and the analysis chamber (base pressure ~5× $10^{-9}$ mbar) were used. Several sputter–anneal cycles (1 keV Ar+) at 650 K combined with oxygen treatments ($p_{O2}$ ~ 1 × $10^{-7}$ mbar, 5 min at 650 K) were performed in the preparation chamber, where the sample can be heated by radiative or e-beam heating using a W filament in close proximity to the backside of the sample plate. Sample heating in the analysis chamber is performed by irradiating the backside of the sample plate with a fibre-coupled infrared laser. After transfer to the analysis chamber (via the transfer chamber), the sample cleanliness was checked at a sample temperature of 650 K. Residual carbon was removed by dosing O$_2$ ($p \leq 1 \times 10^{-7}$ mbar) at 650 K until the C1s region showed a carbon-free surface. Excess O$_{ad}$ was then removed by dosing H$_2$ ($p \leq 1 \times 10^{-5}$ mbar) at 650 K. Both UHV and NAP XPS can be performed in the analysis chamber. During measurements, the sample surface is ~0.3 mm away from the entrance of the differentially pumped Scienta HIPP3 electron energy analyser. After preparation in UHV conditions, the NAP cell, with a volume of ~500 mL, can be closed and local pressures up to 30 mbar can be introduced while the pressure in the main chamber remains <5 × $10^{-5}$ mbar.

Two stainless steel strips were used to clamp the sample onto a stainless steel flag-style sample holder. The sample temperature was measured using a K-type thermocouple, spotwelded to the sample plate below the Co(0001) sample. A photon energy of 420 eV was used to record C1s spectra during H$_2$ exposure and heating in H$_2$. Owing to the extremely high photon flux generated by MAX IV, acquisition of a single C1s spectrum, in the presence of H$_2$ (g), takes around 30 s. Spectra recorded with 800 eV of a CO-covered surface serve as a quantitative reference to determine the coverages of C$_x$H$_y$ and CO$_{ad}$.

Ethylene (AGA, 99.95) and O$_2$ (AGA, 99.999%) were only dosed at UHV pressures and used without further purification. The H$_2$ (AGA 99.998%) first passed over a Pall GLPSIPVMM4 filter followed by a cold trap held at liquid nitrogen temperature prior to entering the NAP cell via a mass flow controller (MKS GF120). The experiments described in the main text reveal a small contamination with CO. This could be due to a residual contamination in the gas or instead from (H$_2$-induced) desorption from the walls of the gas dosing system or those of the vacuum chamber. A trace amount of sulfur was detected by XPS after prolonged exposure to H$_2$.

**TPRS, LEED and RAIRS in UHV.** The TPRS, LEED and RAIRS experiments were performed in a separate UHV set-up with a base pressure of ~5 × $10^{-10}$ mbar. Here the sample was clamped between the two legs of a 0.5-mm-thick U-shaped tungsten wire that is in thermal contact with a liquid nitrogen reservoir so that a sample temperature of ~95 K can be reached. The sample was heated by passing a direct current through the W support wire, and the temperature was measured using a K-type thermocouple, spotwelded to the backside of the sample. During the TPRS experiments, the sample is placed ~5 mm away from the 5-mm-wide aperture of the (separately pumped) mass spectrometer compartment. This arrangement effectively eliminates peaks due desorption from other parts of the sample holder during heating. The known 0.5 ML H$_{ad}$-covered surface[30,37,58] was used to determine the quantity of H$_2$ produced during C$_x$H$_y$ decomposition. Using the

known C/H ratio in the precursor molecule, this can be translated to ML coverages of C$_x$H$_y$. A mass balance was used to quantify molecular desorption of ethene: H$_2$ desorption shows that 0.12 ML decomposes during heating in vacuum. Decomposition is suppressed down to 0.02 ML when the same C$_2$H$_4$ layer is heated in CO. The difference between C$_2$H$_4$ desorption in vacuum and in CO must therefore be equal to 0.1 ML, and with this reference point other coverages could be determined.

Infrared absorption spectra were recorded using a Perkin Elmer Frontier spectrometer. After leaving the spectrometer, the (p-polarized) light travels through a compartment with custom-made optics that focus the beam onto the 8 mm disc-shaped sample. The light enters the vacuum chamber through a KBr window and is reflected off the sample surface. The angle of incidence is 15° with respect to the surface plane. After reflection, the light leaves the vacuum chamber through another KBr window after which it is focused onto a liquid nitrogen-cooled MCT detector. All parts of the beam path that are at atmospheric pressure are flushed with dry N$_2$ to eliminate signals from CO$_2$ (g) and H$_2$O (g) from the spectra. The spectrum obtained by reflection from a clean Co(0001) sample was subtracted from all spectra shown here, and in addition to this, a spline background was used to eliminate changes of the background resulting from sample heating. All spectra were measured with a resolution of 4 cm$^{-1}$ and a step size of 0.5 cm$^{-1}$. Thirty scans were averaged for each point in the TP-RAIRS experiments (Fig. 3a). The high-quality spectra shown in Fig. 3b are the result of averaging 512 individual scans, recorded at 95 K to minimize thermal broadening. By removing the CO pressure at the annealing temperature ensures that the CO coverage and site occupation for the high-quality spectra is identical to the coverage at the annealing temperature. We found that the intensity of the absorption peak due to the symmetric C-H bending mode of -CH$_3$ in adsorbed 2-butyne, at 1365 cm$^{-1}$, is very low. Since it appeared that cooling in CO led to a somewhat higher intensity of this band, we show the spectrum after cooling in CO in Fig. 2b instead of the cooling in vacuum spectrum where the 1365 cm$^{-1}$ band is more difficult to distinguish from the noise.

A liquid nitrogen trap was used to further purify the CO (CK Specialty Gases Ltd., 99.97%) and H$_2$ (CK Specialty Gases Ltd., 99.999%) used during the TPRS, LEED and RAIRS experiments. Ethylene (Messer, 99.95%) was used without further purification. 2-Butyne (Sigma-Aldrich, 99%) used in reference experiments was degassed by several pump–freeze–thaw cycles prior to use.

## Data availability
The data that support the findings of this study are available within the paper and its Supplementary Information, and all data are available from the authors on reasonable request.

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

## Acknowledgements

This work has been carried out as part of the SynCat@DIFFER programme between the Dutch Institute for Fundamental Energy Research (DIFFER), Eindhoven university of Technology (TU/e) and Syngaschem BV and is funded jointly by the Netherlands Organization for Scientific Research (NWO) and Syngaschem BV (project number 731.016.301). We acknowledge ELETTRA, the European Synchrotron light source in Trieste, Italy (proposal 20180250), and MAX IV, the Swedish National Laboratory for research using X-rays, Lund, Sweden (proposal 20180237) for provision of beamtime. The staff at the SuperESCA (ELETTRA) and HIPPIE (MAX IV) beamlines are acknowledged for their excellent support. We acknowledge the technical support from the technical support staff at the DIFFER Institute. Professor Dr. M.C.M. van de Sanden (DIFFER) is acknowledged for commenting on the manuscript and providing valuable suggestions. Syngaschem BV gratefully acknowledges substantial funding from Synfuels China Technology Co. Ltd.

## Author contributions

The experiments at the SuperESCA beamline of ELETTRA, Trieste were performed by D.S., D.G.R., M.A.G. and C.J.W. NAP-XPS measurements at the HIPPIE beamline of Max IV, Lund were performed by D.S., D.G.R., H.O.A.F. and C.J.W. TPRS, LEED and RAIRS experiments were performed by C.J.W. and supporting TPRS measurements were performed by D.S. The work was conceived and designed by C.J.W. who also wrote the principal draft of the paper with contributions from D.S., D.G.R., M.A.G. and H.O.A.F., followed by close review and editing by J.W.N.

## Competing interests

The authors declare no competing interests.
