## [Peer Review File · Nature Communications]

Reviewers' comments:

Reviewer #1 (Remarks to the Author):

This paper uses a lot of characterization technologies to detect the C-C bond formation process details in Fischer-Tropsch synthesis (FTS) at the initial stage. CO is found to stabilize ethynyl. To my knowledge, a lot of C-C chain propagation mechanisms are reported until now, including some ones including oxygenated compounds as intermediates, just as this paper. But the conclusions here are not proper for all metallic surfaces. Most of the data are from Co(0001).

(1) Bridge-type adsorbed CO on flat Co surface is the main active site. But CO adsorbed at the edge and corner of Cobalt particle also has enough activity to contribute to CO hydrogenation. Please consider the role of these CO molecules.

(2) H₂O partial pressure is rather high at the FTS Co catalyst surface. It controls and influences the surface dynamics. Please consider this effect.

(3) Co is partially oxidized during FTS reaction due to H₂O and CO. ML of CO coverage will be controlled by Co reduction degree.

(4) I suggest the authors to add isotope experiments using labelled C(13) to catch C atom changing route.

(5) Using CH₃I to tune CH₃ surface coverage on Co surface is not reliable as residual I will alter Co electronic state dramatically.

(6) CO and H₂ can attack ethylene readily to start hydroformylation reaction, forming C₃ aldehyde. So of course CO can stabilize C₂ species (ethylene, ethynyl, etc).

In my opinion, catalysis journal or surface science journal would be more suitable for this manuscript.

Reviewer #2 (Remarks to the Author):

Weststrate and co-workers report another very nice and careful study of elementary reactions involved in Fischer-Tropsch chain growth. The study convincingly demonstrates the effect of the CO coverage on the nature of the growing chains, and directly observes C-C coupling on Co terraces. These results are very important for the field of Fischer-Tropsch synthesis and should certainly be published.

A minor concern is the overlap with reference 7, where the effect of CO coverage on the stability of the growing chain was first demonstrated, but I guess this discussion is necessary for the current story. Maybe some of the overlap could be reduced by shortening certain parts.

The authors speculate on the reason for the effect of the CO coverage, but do not provide a fully convincing reason. Attractive interactions between ethynyl and CO could be important, but what would be the nature of this attraction – dipole interactions? Relatively stronger repulsive interactions with acetylene (which has a larger footprint on the surface) seem more important; repulsion between H and CO seems less important.

The connection with NMR on p 3 is nice, but farfetched. NMR shifts are caused by changes in the electron distribution in the probed nucleus by substituent effects, while shifts in the XPS peak are caused by energy-loss by the photo-electron to vibrational modes. The authors should not overstate the link between NMR and XPS.

The difference between the measured barrier of 60 kJ/mol and computed barriers could be due to coverage effects.

Some authors suggest that chains grow at defect sites. Maybe the authors could comment on the active sites for chain growth.

Reply to reviewers

General comments

We thank both reviewers for their valuable comments that helped to improve the quality of our article. We regret that the first reviewer focuses predominantly on mechanistic interpretations and does not seem to have considered implications of the present experiments very much, while such experimental results are in fact precious in the FTS field, where much thinking is based on molecular modeling these days. It appears that it was not very clear to reviewer one that the C-C bond formation that we observe in our experiments proceeds via a reaction of oxygen-free intermediates. This means that CO is not directly involved but acts as an important spectator. We have changed the title and abstract to better emphasise the key message of the article, the mechanism of C-C bond formation, and added the text provided below to the introduction to better to clarify this:

New title: New Light on an Old Catalyst: Mechanistic insight into Carbon-Carbon bond formation on Cobalt relevant to Fischer-Tropsch Synthesis

Text added (p2, last paragraph of the introduction):

We here use a Co(0001) model catalyst to study how $C_2H_{x,ad}$ species react to form a new C-C bond under FTS-like conditions, that is, in the presence of co-adsorbed hydrogen and CO_{ad} . Using this approach, we find that C-C bond formation is promoted by CO spectators which stabilize the alkylidyne intermediate needed for this reaction. This finding can rationalize why CO promotes alkene dimerization on cobalt catalysts and reveals the hidden role of CO as promoter of chain growth during Fischer-Tropsch Synthesis on supported cobalt catalysts.

Reviewer 1

Reviewer remark:

This paper uses a lot of characterization technologies to detect the C-C bond formation process details in Fischer-Tropsch synthesis (FTS) at the initial stage. CO is found to stabilize ethylidyne.

- (1) *To my knowledge, a lot of C-C chain propagation mechanisms are reported until now, including some ones including oxygenated compounds as intermediates, just as this paper.*

Reply:

We would like to emphasise that mechanisms proposed in literature are often inferred from indirect observations, or based on theory calculations. In contrast to this, our mechanistic proposal is based on experimental observation of key reaction steps, that is, C-C bond formation, (CO-promoted) alkyne-alkylidyne conversion and slow hydrogenation of alkyne adsorbates. We in other words provide direct experimental evidence that these steps are possible, and feasible under FTS conditions in view of the temperatures where they occur and the high surface coverage that was present during our model experiments to simulate the high coverage expected under FTS conditions.

The reviewer statement "*including oxygenated compounds as intermediates, just as this paper*" suggests that the reviewer is under the impression that the paper describes oxygenate formation. This is not the case. In our paper we show how the presence of CO has an important impact on how

hydrocarbon adsorbates react on a cobalt surface. But it is important to note that we in fact find no indications for the involvement of oxygen-containing intermediates that would be formed if CO reacts with C_xH_y adsorbates. The supplementary information that accompanied the originally submitted manuscript already contained a detailed discussion where we exclude the possibility that oxygenates are formed at any stage in the experiment. However, the link to this part of the supplementary information was not very clear in the main article. We therefore added the following text to the discussion section to emphasize this point:

Text added (p 12, end of page):

The mechanism observed in our experiments is of the carbide type, and, although CO is promoting chain growth by stabilizing the intermediates in the correct form, it does not participate in the C-C bond forming reaction. A detailed analysis of the O1s spectra provided in the supplementary information (Fig. S10) shows that the formation of oxygen-containing intermediates during the experiments can be excluded.

Reviewer remark:

(2) But the conclusions here are not proper for all metallic surfaces. Most of the data are from Co(0001).

Reply:

We agree with the reviewer that the experimental data was obtained on the flat, close-packed cobalt surface and the proposed mechanism may not be the same at different surface sites. We therefore added a section in the discussion where we rely on insights from theory and experiments on nickel surfaces to discuss chain growth at step edge sites. We added the following paragraph to the discussion to address this question:

Text added (p13):

The experimentally observed surface reactions presented here show that C-C bond formation via alkylidyne coupling is favourable on the close-packed terraces that constitute ~60% of the surface of an (fcc-)Co nanoparticle^{4,42}. The methylidyne monomers required for chain growth during FTS can either be supplied via H-assisted routes on the terrace or via spill-over from adjacent step sites that are active for CO dissociation^{18,51} and surround the nm-sized close-packed terrace exposed by an fcc-Co nanoparticle⁴. An fcc-Co nanoparticle also exposes (100) facets as well as step and kink sites⁵², and the role of the different structural elements remains a topic of debate⁵³. Given that our experiments only speak about C_xH_y reactivity on close-packed terraces we can only provide a qualitative consideration about chain growth on other sites. Since the relative stabilities of C_xH_y adsorbates depends on the surface structure chain growth may proceed via a different pathway on other parts of the catalyst surface. In fact, experiments on stepped Ni(111) show that step sites favour C-C bond scission rather than C-C bond formation^{54,55}, and C-C bond scission was reported on Ni(100) at temperatures below 300 K⁵⁶. This is in line with computational work on cobalt, which shows that 2 C₁H_x adsorbates are more stable than C₂H_x adsorbates on both Co(100) and stepped Co(211), while acetylene and ethylidyne are instead significantly more stable than 2 CH species on Co(111)⁴². In other words, C-C bond scission is favoured on steps and (100) facets whereas C-C bond formation is favoured on terrace sites. This means that growth of long chains at step edges can only occur when exothermic C-C bond breaking is strongly suppressed, e.g. by a local high coverage at the step site. The concentration of carbon is indeed expected to be high on step sites due to their high affinity for carbon^{42,57}. Chain growth requires growing chains to physically meet with a monomer and since the growing chain concentration is expected to be small⁶ the nearest monomer is most probably generated at some distance from the growth intermediate. A high local coverage

at step edges poses a significant barrier for C_xH_y diffusion along the 1D step edge, making it difficult for a monomer to reach a growing chain that is located a few sites away. Thus, due to limited mobility along the highly covered step sites only those monomers formed at a small ensemble of step sites will contribute to a single chain, whereas monomers produced at step sites far away may not be able to reach the growing chain and make methane instead. Instead, spill-over of monomers to the adjacent (2D) terrace circumvents blockages of strongly adsorbed species at step sites so that monomers and growing chains can more easily diffuse over a larger distance and monomers produced by different active sites located at the edges of a single terrace can all be incorporated into the same product chain.

Text added (p14, underlined part was added to the conclusions):

For FTS we propose that C-C bonds form on the close-packed facets via coupling of a long chain alkylidyne with a methylidyne monomer to form a 1-alkyne adsorbate.

Reviewer remark:

(3) Bridge-type adsorbed CO on flat Co surface is the main active site. But CO adsorbed at the edge and corner of Cobalt particle also has enough activity to contribute to CO hydrogenation. Please consider the role of these CO molecules.

Reply:

We do not understand the remark of the reviewer. The IR spectra in fact exclude that bridge sites are occupied throughout the experiment. The reviewer may refer to the issue of C-O bond scission, an essential step in the overall FTS process. However, our work concerns the C-C bond formation step required for chain growth. Since the issue of C-O bond scission and the formation of monomers is discussed by many authors in other publications we refer to those papers for more information about the location where C-O bond breaking takes place. We again stress that we do not have any indication that CO reacts in the C-C bond forming step on the flat surface. We have added the following sentence to the discussion to clarify where the monomers required for chain growth may come from:

Text added (p13, top):

The methylidyne monomers required for chain growth during FTS can either be supplied via H-assisted routes on the terrace or via spill-over from adjacent step sites that are active for CO dissociation^{18,51}

Reviewer remark:

(4) H₂O partial pressure is rather high at the FTS Co catalyst surface. It controls and influences the surface dynamics. Please consider this effect.

Reply:

This statement is not at all undisputed in the FTS community. We agree with the reviewer that applied FTS is operated at high conversions so that the partial pressure of the H₂O will be significant. However, it is also known that Fischer-Tropsch synthesis does not require H₂O to work, since long chains can also be produced when the conversion is low. The real catalyst system is too complex to elucidate

molecular details about the chain growth mechanism and that is why we use a simplified model system to identify the adsorbates that are reactive for C-C bond formation and are at the same time stable enough to stay on the surface for a significant amount of time under reaction conditions. Adding water would be very difficult experimentally and would increase the complexity of the experiments even further. As the presence of water does not seem to be essential for chain growth we leave the influence of water as a topic for further study. We added the following section to the introduction to address this comment:

Text added (p2, 2nd paragraph):

Water is the main product of FTS on a molar basis, and the high conversion levels reached during industrial operation lead to a water partial pressure in the order of a few bars. However, since chain growth also occurs under low conversion conditions where the H₂O partial pressure is low, the presence of H₂O does not appear to be essential to the chain growth mechanism and was therefore omitted from our study. Moreover, surface science studies show that water adsorbs much weaker on Co(0001) than both CO and hydrogen¹³, so the H₂O surface coverage under reaction conditions is expected to be low even when the H₂O partial pressure is comparable to that of CO and H₂.

Reviewer remark:

(5) Co is partially oxidized during FTS reaction due to H₂O and CO. ML of CO coverage will be controlled by Co reduction degree.

Reply:

This statement is a topic of discussion, and is in particular complicated to investigate experimentally since the use of oxidic supports in applied FTS most often leads to incomplete reduction of the fresh catalyst so that oxide phases are always present throughout the FTS reaction. Propensity for oxidation is also dependent on particle size, where the small particles, those that show lower activity and selectivity in FTS are more prone to oxidation than large particles that show good FTS performance (<https://doi.org/10.1021/jp045136o>). Oxidation is typically considered as a deactivation mechanism (<https://doi.org/10.1016/j.cattod.2007.02.032>), and, as mentioned in the reply to remark #4, chain growth in FTS does not require H₂O, since it can also occur at low conversions already. Moreover, in model-type studies carbon-supported Co was employed (<https://doi.org/10.1021/ja058282w>) and full reduction of Co was achieved. No significant oxidation of either bulk or surface was found, indicating that metallic cobalt is the active phase. We note in addition that metallic single crystals have shown activity for FTS and are therefore good model systems to study for FTS (<https://doi.org/10.1038/s41929-019-0360-1>, [https://doi.org/10.1016/0039-6028\(91\)90091-6](https://doi.org/10.1016/0039-6028(91)90091-6)). Finally, we note that we are not aware of mechanistic studies that oxidized cobalt rather metallic cobalt as the active phase.

Text added (p2, 2nd paragraph):

An in-situ EXAFS study of cobalt supported on a carbon nanofiber support shows that neither bulk oxidation nor substantial surface oxidation occurs on cobalt during FTS¹⁴. Furthermore, cobalt single crystals were found to be active for Fischer-Tropsch synthesis^{11,15-18}, and the turn-over frequencies reported are similar to those found for supported catalysts. This confirms that metallic cobalt is the active phase for chain growth and that insights from single crystal studies are of direct relevance for fundamental understanding of FTS.

Reviewer remark:

(6) I suggest the authors to add isotope experiments using labelled C(13) to catch C atom changing route.

Reply:

We cannot see how using C13 labelled compounds will provide additional information. In our work we focus on elementary reaction steps that occur on the surface of the catalyst in which only the C_xH_y adsorbates (not the CO) react and the products remain adsorbed on the surface. We use XPS to identify the intermediates, but XPS cannot distinguish different isotopes so we would not see any difference by using C13-labeled species in our spectra. The mass spectrometer would be able to detect isotope incorporation, but since the products of the reaction remain adsorbed and we see only hydrogen produced in the UHV experiment using C13 labelled compound will not provide relevant information.

Reviewer remark:

(7) Using CH3I to tune CH3 surface coverage on Co surface is not reliable as residual I will alter Co electronic state dramatically.

Reply:

In the process of shortening the manuscript we have removed the part that speaks about the experiments where CH3I was used. Results were therefore also eliminated from the supplementary information.

Reviewer remark:

(8) CO and H2 can attack ethylene readily to start hydroformylation reaction, forming C3 aldehyde. So of course CO can stabilize C2 species (ethylene, ethylidyne, etc).

Reply:

The hydroformylation reaction is known to be catalysed by cobalt carbonyl complexes in homogeneous catalysis but it is not commonly believed that this is a major reaction on a heterogeneous cobalt catalyst. And in the hydroformylation reaction the C-C bond is formed by CO insertion, and an aldehyde product is formed. We repeat that we exclude a reaction between CO and C_xH_y.

Reviewer #2:

Reviewer remark:

(9) Weststrate and co-workers report another very nice and careful study of elementary reactions involved in Fischer-Tropsch chain growth. The study convincingly demonstrates the effect of the CO coverage on the nature of the growing chains, and directly observes C-C coupling on Co terraces. These results are very important for the field of Fischer-Tropsch synthesis and should certainly be published.

Reply:

We thank the reviewer for the positive comments on our work and the positive recommendation.

Reviewer remark:

(10) A minor concern is the overlap with reference 7, where the effect of CO coverage on the stability of the growing chain was first demonstrated, but I guess this discussion is necessary for the current story. Maybe some of the overlap could be reduced by shortening certain parts.

Reply:

We agree with the reviewer that the work builds on previous work and we considered it useful to briefly discuss key points of earlier publications to facilitate comparison with the new results presented in the present article. The main data consists of XPS measurements in UHV in presence of CO, and XPS at near-ambient pressures and such data have not been reported before. We added XPS results in the absence of CO which was previously discussed in detail elsewhere and here serves as a reference to highlight the large differences induced by the presence of CO. Likewise, the TPRS and IR measurements included in the present manuscript corroborate earlier published TPRS and IR work and in the present work serve as supporting evidence for the XPS data. The conclusions reached in the present paper to some extent overlap with those presented in our earlier paper (<https://doi.org/10.1021/acscatal.8b02743>) There we used IR data, which provides qualitative information about the species formed, in combination with TPRS to demonstrate the effect of CO coverage on the growing chain for the first time. The experimental data presented in our earlier paper is not easy to understand and difficult to explain convincingly. A reviewer comment for that article says it all: "The interpretations of the vibrational data seems to be quite reasonable." The formation of an ordered ethynidyne/CO overlayer was not reported before and it is an important finding that provides details on the origin of the CO affects CxHy stability changes.

Although we speculated on the possibility of C-C bond formation in our previous work the experimental evidence for this was very weak. The key result of the present work is the in-situ observation of a C-C bond forming reaction on a cobalt surface, which is facilitated by CO spectators. The XPS data presented here provides us with a straightforward identification of all intermediates throughout the reaction sequence that leads to C-C bond formation and the quantitative evaluation provides us with the concentration of the reactants at each temperature so that we can follow two surface reactions as they proceed and determine the barrier associated with both. The clear advantage of XPS over RAIRS and TPRS that all adsorbates can be directly detected (except H_{ad}, which could be indirectly derived from TPRS) also makes it possible for us to rigorously exclude that CO insertion takes place at any stage in our experiments.

We for the first time report measurements at near-ambient pressures, non-trivial experiments that convey an important message, namely that the findings at UHV also apply to higher pressures. They in other words show that the pressure gap does not seem to play a large role and that our findings are relevant for applied catalysis. The NAP-XPS also allowed us to follow a reaction that we were never able to detect in our typical UHV approach: the barrier associated with hydrogenation of adsorbed alkynes to (gas phase) alkene product, the last and in fact rate-determining step in ethylene dimerization that closes the catalytic cycle. It is also very important to have experimental information about a slow termination reaction in FTS as the rate of termination is an important determining factor in the selectivity of the reaction. Finally, we would like to stress that all data presented here in some cases reproduces earlier published work but in all cases concerns new measurements that have not been published before.

Reviewer remark:

(11) The authors speculate on the reason for the effect of the CO coverage, but do not provide a fully convincing reason. Attractive interactions between ethynidyne and CO could be important, but what would the nature of this attraction – dipole interactions? Relatively stronger repulsive interactions with acetylene (which has a larger footprint on the surface) seem more important; repulsion between H and CO seems less important.

Reply:

Following the reviewer suggestion we have expanded the discussion on this topic. The driving force for the mixed ethylidyne-CO layer was investigated previously in more detail on Rh(111). We briefly mention the main conclusion of the work and provide the reader with a reference for further reading. We have also added considerations about the footprint of ethylidyne and acetylene to the discussion. The text added in the discussion section is provided below:

Text added (p10, end of first paragraph):

Mate et al.⁴³ attributed the ordering of mixed ethylidyne/CO layers on Rh(111) to favourable dipole-dipole interactions due to the oppositely oriented dipoles of CO and ethylidyne. A third contributing factor is that a high surface coverage favours species that occupy less space on the surface²¹. This makes acetylene hydrogenation to ethylidyne favourable, firstly because it incorporates a surface hydrogen into the C_xH_y adsorbate, and secondly, the footprint of ethylidyne is expected to be lower than that of acetylene since the former adsorbs through only one carbon atom.

Reviewer remark:

(12) The connection with NMR on p 3 is nice, but farfetched. NMR shifts are caused by changes in the electron distribution in the probed nucleus by substituent effects, while shifts in the XPS peak are caused by energy-loss by the photo-electron to vibrational modes. The authors should not overstate the link between NMR and XPS.

Reply:

We have removed this connection from the article

Reviewer remark:

(13) The difference between the measured barrier of 60 kJ/mol and computed barriers could be due to coverage effects.

Reply:

We rephrased the paragraph that discusses the barrier of acetylene to ethylidyne to improve the discussion and to accommodate the comment made by the reviewer. The new part is provided below:

Revised text (p9, center):

The conversion of acetylene to ethylidyne requires the addition of two hydrogen atoms to one end of the molecule and abstraction of a hydrogen atom from the other end. Our analysis yields an overall barrier height of only 60 ± 6 kJ·mol⁻¹, far below the computed barriers >100 kJ·mol⁻¹ as predicted by DFT starting with either acetylene hydrogenation or acetylene dehydrogenation as a first step⁴⁰⁻⁴². This indicates that the high CO coverage may also lower the reaction barriers involved in the conversion of acetylene.

Reviewer remark:

(14) Some authors suggest that chains grow at defect sites. Maybe the authors could comment on the active sites for chain growth.

Reply:

We considered this remark to be in essence similar to the remark #2 made by reviewer #1 and the text added in response is provided there. At this point we want to make some additional remarks about our considerations regarding this issue:

Our experiments on a model system provides direct experimental evidence showing that C-C bond formation chain growth is possible and in fact favourable on the close-packed surface, and even promoted by the presence of CO. We consider this our main and most important message that we want to bring to the attention of the FTS community. It is certainly fair to ask the question how these very clear findings on a simple model system fits into the bigger picture of applied FTS, where a catalyst particle expose a number of surface sites including the step edge sites proposed by other authors. The complexity of addressing this remark in the present manuscript is that our data does not contain any information about the reactivity of other sites, so the discussion has to be based on previous publications. And since different sites are present in close proximity one in fact has to consider the reactivity of different sites and also the possibility of spill-over from one site to the other, making the situation and the discussion thereof very complex.

In light of this complexity we included the following elements in a short discussion on the topic:

- (i) We start by describing a scenario in which chain growth occurs on the terraces of a nanoparticle. We emphasize that the chain growth mechanism on terrace sites would be valid independent on whether the CH monomer is made on the terrace via H-assisted pathways as suggested by some authors, or at the step edge sites that surround close-packed terraces on a nanoparticle and are active for direct C-O bond scission, as suggested by other authors.
- (ii) We mention that other sites are exposed on a real nanoparticle as well, and that chain growth mechanism may be different there. However, our data does not provide direct information and we have to rely on other experimental and theoretical work to get more insights.
- (iii) We then mention that experiments and theory on step sites show that C-C bond formation on step sites is endothermic, a large difference with the situation on the close packed surface described in the article, where C-C bond formation was found to be exothermic.
- (iv) We then go on to describe some aspects of a scenario where chain growth happens exclusively at step edges. We provide a few qualitative conceptual complications from a surface science perspective that make this scenario in our view somewhat complicated: the preference for C-C bond breaking at step edges is working against chain growth, but does not make it impossible. But it means that chain growth can only happen when the reverse reaction, C-C bond breaking, is suppressed by a high coverage at step edges so that there are no vacancies available to facilitate C-C bond breaking. But at the same time diffusion of reactants is important for chain growth, since it requires growing chains to meet with monomers. Here the 1D nature of step edges in combination with a local high coverage makes diffusion difficult, since any adsorbate encountered during diffusion of monomers or growing chains while they diffuse along the step sites will cause a blockage that cannot be overcome unless the adsorbate that blocks the path reacts away or if instead the growing chain or monomer transfers to the adjacent, 2D terrace where the 2D nature makes diffusion more easy and blocking co-adsorbates can be more easily circumvented.

Text added (p13):

See comment #2, reviewer #1

REVIEWERS' COMMENTS:

Reviewer #1 (Remarks to the Author):

This resubmitted manuscript is improved and most of my previous questions are well answered. I still have several questions and hope the authors to consider, or add into the text.

(1) The authors declare that no reaction exists between CO and C_xH_y. What is the intrinsic interaction in the stabilization effect from CO with high coverage to surface C_xH_y?

(2) Ethylene dimerization to form 2-butylene, or methylidyne insertion to ethylidyne, to extend carbon chain, was observed at very low temperature (300K here). The real Fischer-Tropsch synthesis works at 220C to 240C with high partial pressure of H₂ from syngas. The reported phenomenon here (C-C chain growth of unsaturated hydrocarbon species) should be very difficult to appear as these species are hydrogenated quickly.

Reviewer #2 (Remarks to the Author):

The authors have addressed my comments and improved the manuscript. I remain convinced that these are important results that should certainly be published.

Rebuttal to additional Reviewers comment:

Reviewer #1 (Remarks to the Author):

This resubmitted manuscript is improved and most of my previous questions are well answered. I still have several questions and hope the authors to consider, or add into the text.

(1) The authors declare that no reaction exists between CO and C_xH_y. What is the intrinsic interaction in the stabilization effect from CO with high coverage to surface C_xH_y?

We refer the reviewer to the second paragraph of the discussion, where we provide considerations regarding the various factors that contribute to the observed stabilization of ethylidyne. The manuscript contains the following text (discussion, 2nd paragraph) to discuss this topic: “The experiments show that CO causes conversion of acetylene to ethylidyne, and the driving force for this must be that ethylidyne becomes more stable than acetylene when CO is present on the surface^{32,40}. Two experimental observations give more insight into *why* ethylidyne is preferred when CO is present: (i) repulsive interactions between H_{ad} and CO_{ad} destabilize H_{ad} [Fig. 2(b)], thereby *destabilizing* one of the reactants. (ii) The formation of ordered CO_{hollow}/ethylidyne layer *islands* with a local high coverage is driven by *attractive* interactions, indicating that the ethylidyne reaction product is *stabilized* by CO_{ad} [Fig. 3(c)]. Mate et al.⁴² attributed the ordering of mixed ethylidyne/CO layers on Rh(111) to favourable dipole-dipole interactions due to the oppositely oriented dipoles of CO and ethylidyne. A third contributing factor is that a high surface coverage favours species that occupy less space on the surface²¹. This makes acetylene hydrogenation to ethylidyne favourable, firstly because it incorporates a surface hydrogen into the C_xH_y adsorbate, and secondly, the footprint of ethylidyne is expected to be lower than that of acetylene since the former adsorbs through only one carbon atom.”

Since this information was already provided in the article we did not make any modifications to the manuscript.

(2) Ethylene dimerization to form 2-butyne, or methylidyne insertion to ethylidyne, to extend carbon chain, was observed at very low temperature (300K here). The real Fischer-Tropsch synthesis works at 220C to 240C with high partial pressure of H₂ from syngas. The reported phenomenon here (C-C chain growth of unsaturated hydrocarbon species) should be very difficult to appear as these species are hydrogenated quickly.

The rate at which surface intermediates are hydrogenated is indeed an important factor that needs to be taken into account when considering chain growth mechanisms. In the 6th paragraph of the discussion we discuss that chain growth intermediates need to have a high stability, and we argue that only the most stable C_xH_y intermediates which are most resistant to hydrogenation qualify as growth intermediates. The fact that we observe acetylene, ethylidyne and 2-butyne in our experiments suggests that alkynes and alkylidynes are among the most stable C_xH_y species, a notion that is confirmed by theory calculations which identify alkynes as the most stable C_xH_y adsorbates on Co(0001). The difficulty to hydrogenate adsorbed alkynes is evident from the high temperature required to hydrogenate 2-butyne, even when heated in pure H₂.

The manuscript (discussion, paragraph 6) contains the following text to address this point “Such slow growth on the second time scale can only involve very stable growth intermediates that have difficulty leaving the surface. Our work shows that alkynes and alkylidynes are the most stable C_xH_y adsorbates

possible, and the NAP-XPS experiment (fig. 5) confirms that alkyne hydrogenation is a slow reaction that only proceeds at a significant rate above 370 K. Their high stability and long lifetimes thus make alkynes and alkylidynes feasible intermediates for chain growth under FTS conditions.”.

Reviewer #2 (Remarks to the Author):

The authors have addressed my comments and improved the manuscript. I remain convinced that these are important results that should certainly be published.

We thank the reviewer for the positive comments on the improved manuscript.